# Novel Harmonization Method for Multi-Centric Radiomic Studies in Non-Small Cell Lung Cancer

**Marco Bertolini** [1] , **Valeria Trojani** [1,*] , **Andrea Botti** [1] , **Noemi Cucurachi** [1] , **Marco Galaverni** [2] ,
**Salvatore Cozzi** [3] , **Paolo Borghetti** [4] , **Salvatore La Mattina** [4] , **Edoardo Pastorello** [4] , **Michele Avanzo** [5] ,
**Alberto Revelant** [6] , **Matteo Sepulcri** [7] , **Chiara Paronetto** [7] , **Stefano Ursino** [8] , **Giulia Malfatti** [8] ,
**Niccolò Giaj-Levra** [9] , **Lorenzo Falcinelli** [10] , **Cinzia Iotti** [3] , **Mauro Iori** [1] **and Patrizia Ciammella** [3]

1. S.C. Fisica Medica, Azienda USL-IRCCS di Reggio Emilia, 42124 Reggio Emilia, Italy;
   marco.bertolini@ausl.re.it (M.B.); andrea.botti@ausl.re.it (A.B.); noemi.cucurachi@ausl.re.it (N.C.);
   mauro.iori@ausl.re.it (M.I.)
2. S.C. Radioterapia, Azienda Ospedaliero-Universitaria Maggiore, 43126 Parma, Italy; mgalaverni@ao.pr.it
3. S.C. Radioterapia, Azienda USL-IRCCS di Reggio Emilia, 42124 Reggio Emilia, Italy;
   salvatore.cozzi@ausl.re.it (S.C.); cinzia.iotti@ausl.re.it (C.I.); patrizia.ciammella@ausl.re.it (P.C.)
4. Department of Radiation Oncology, University and Spedali Civili Hospital, 25123 Brescia, Italy;
   paolo.borghetti@asst-spedalicivili.it (P.B.); s.lamattina@unibs.it (S.L.M.); e.pastorello@unibs.it (E.P.)
5. Medical Physics Department, Centro di Riferimento Oncologico di Aviano (CRO) IRCCS, 33081 Aviano, Italy;
   mavanzo@cro.it
6. Radiation Oncology Department, Centro di Riferimento Oncologico di Aviano (CRO) IRCCS, 33081 Aviano,
   Italy; alberto.revelant@cro.it
7. Radiotherapy, Veneto Institute of Oncology IOV—IRCCS, 35128 Padova, Italy;
   matteo.sepulcri@iov.veneto.it (M.S.); chiara.paronetto@iov.veneto.it (C.P.)
8. Department of Radiation Oncology, Santa Chiara University Hospital, 56100 Pisa, Italy;
   stefano.ursino@med.unipi.it (S.U.); giulia.malfatti@med.unipi.it (G.M.)
9. Department of Radiation Oncology, IRCCS Sacro Cuore Don Calabria Hospital Negrar, 37024 Verona, Italy;
   niccolo.giajlevra@sacrocuore.it
10. Radiation Oncology Section, S. Maria della Misericordia Hospital, 06129 Perugia, Italy;
    lorenzo.falcinelli@ospedale.perugia.it
* Correspondence: valeria.trojani@ausl.re.it

**Abstract:** The purpose of this multi-centric work was to investigate the relationship between radiomic features extracted from pre-treatment computed tomography (CT), positron emission tomography (PET) imaging, and clinical outcomes for stereotactic body radiation therapy (SBRT) in early-stage non-small cell lung cancer (NSCLC). One-hundred and seventeen patients who received SBRT for early-stage NSCLC were retrospectively identified from seven Italian centers. The tumor was identified on pre-treatment free-breathing CT and PET images, from which we extracted 3004 quantitative radiomic features. The primary outcome was 24-month progression-free-survival (PFS) based on cancer recurrence (local/non-local) following SBRT. A harmonization technique was proposed for CT features considering lesion and contralateral healthy lung tissues using the LASSO algorithm as a feature selector. Models with harmonized CT features (B models) demonstrated better performances compared to the ones using only original CT features (C models). A linear support vector machine (SVM) with harmonized CT and PET features (A1 model) showed an area under the curve (AUC) of 0.77 (0.63–0.85) for predicting the primary outcome in an external validation cohort. The addition of clinical features did not enhance the model performance. This study provided the basis for validating our novel CT data harmonization strategy, involving delta radiomics. The harmonized radiomic models demonstrated the capability to properly predict patient prognosis.

**Keywords:** imaging biomarkers and radiomics; quantitative imaging/analysis; computed tomography (ct); multi-modality ct-positron emission tomography (pet); machine learning; non-small-cell lung cancer; stereotactic body radiation therapy (sbrt)

## 1. Introduction

Non-small-cell lung cancer (NSCLC) is, overall, the second-most-common cancer and a leading cause of cancer-related death worldwide, despite recent therapeutic advances [1]. Stage I disease represents approximately 25% of the patients receiving diagnoses of NSCLC and accounts for the most curable cohort of the population [2]. Surgery is the gold standard for these patients: lobectomy with hilar and mediastinal lymph node dissection is the preferred approach, given the Lung Cancer Study Group (LCSG) trial results [3]. Instead, sublobar resection has shown inferior local control and a trend toward decreased survival. However, evaluation of sublobar resection in selected patients is currently underway. The historical standard therapy for unresectable early-stage NSCLC was conventionally fractionated radiation therapy (RT) (e.g., 2 Gy per fraction, for a total dose of 54–60 Gy). However, the reported long-term local control (LC; 30–70%) and overall survival (OS;15–30%) rates with this approach are suboptimal [4–6].

Advances in imaging and radiation treatment planning and delivery (e.g., with image guidance and motion management) made the delivery of "ablative doses" of radiation to small targets possible with better results in terms of local control [7–11].

Stereotactic Body radiation therapy (SBRT) has proved to be the first therapeutic option in inoperable stage I NSCLC patients or for those who refuse surgical treatment, with similar rates of local tumor control and overall clinical outcomes [12,13]. Recently, a meta-analysis by Li et al. reported a significant superiority in the local control rate and in 3-year and 5-year OS (54.73% and 29.30 % vs. 39.5 and 27.47) in the SBRT group compared with conventionally fractionated RT [14].

SBRT was reported to have a local control rate in excess of 85% at 3 years [14,15]. Despite consistent clinical outcomes, it is well known that dose fractionation heterogeneity and technical expertise may influence the outcome with SBRT [16–18]. A recent study reported that the factors affecting outcomes after SBRT for early-stage NSCLC are Biological Effective Dose (BED) and tumor size [19].

Radiomics is a recent technique introduced in medicine to describe characteristics of medical images quantitatively. Radiomics belongs to artificial intelligence (AI) applications, but it is based on the calculation of features using well-defined mathematical formulas applied directly to the image pixel values (or to a filtered version of the original images). The mathematical definitions of radiomic features are based on the distribution and the relationship between pixels and voxels in the images' region of interest. The concept behind this method lies in the fact that the human eye cannot appreciate all the characteristics of a medical image. Haralick et al. [20] described how the textural features, highlighting the behavior of gray levels' dependencies, can identify different areas in an image. Later, textural information was proposed as an application in medical imaging [21,22]. The improvement in hardware calculation power made these techniques able to compute a high number of medical imaging biomarkers in an acceptable span of time; those indices should help the physician during the treatment decision task, allowing a personalized care pathway for different patients. However, these biomarkers are not yet ready to be used in oncology without a robust validation or a demonstration of their reliability [23]. Among them, radiomic indices and feature signatures are increasingly present in the panorama of modern scientific literature [24,25]. The main issue and challenge up to date are to understand how to overcome the limits of this approach [26,27].

To date, in the literature, several studies have investigated the ability of radiomics features in the tumor-healthy tissue differentiation task, both for computed tomography (CT) and positron emission tomography (PET) datasets, as described by Chu et al. [28] that used feature values in a random forest classifier for diagnostic purposes. In another study [29], healthy tissues' features were used as additional information for an automatic segmentation algorithm. More recently, feature-extracted CT images were combined with BED values to predict tumor response to SBRT [30].

Despite the great work done to date, to our knowledge, there is still no characterization of the radiomic features' ability to give specific information about healthy tissue compared

to the sick one when machine learning models for prognosis are involved. One of the main challenges in the field of radiomics, which makes its clinical application difficult, is the harmonization of the features to be analyzed.

Our present multicentric work aims to propose a novel concept of harmonizing the CT radiomic signal using a combination derived from both the tumor and the healthy contralateral tissue, to overcome the variability typical in each patient in different conditions (i.e., manufacturer/technical characteristics, acquisition, reconstruction protocol, and different anatomy).

## 2. Materials and Methods

### 2.1. Study Design

This study was a retrospective multicentric work. It was approved by the Area Vasta Emilia Nord (AVEN) Ethics Committee (ID: 817/2018/OSS*/IRCCSRE). The study was also approved by the ethics committees of all the participating institutions; it was performed in accordance with the principles of Good Clinical Practice (GCP) in respect of the ICH GCP guidelines, the ethical principles contained in the Helsinki declaration and its subsequent updates. Each patient gave informed consent for joining the study.

#### 2.1.1. Patient Cohort

Patients who underwent SBRT for histologically proven diagnosis of primary early-stage NSCLC were retrospectively collected from January 2010 to December 2019. A multicenter research project named "TEXture Analysis of PET/CT in lung cancer patients treated with Stereotactic body radiation therapy (TEXAS)" was designed to involve seven Italian Centers.

Inclusion criteria were: (1) histologically proven diagnosis of NSCLC; (2) early-stage T1–T3N0M0 (TNM 7th edition); (3) patients who underwent SBRT, with treatment biological effective dose $BED_{10} \geq 100$ Gy; and (4) age > 18 years.

Exclusion criteria were: (1) lung tumor greater than 7 cm; (2) histologically proven diagnosis of small cell lung cancer or metastasis; (3) previous thoracic irradiation; (4) presence of bone, lymph node, or visceral metastatic lesions; (5) patients with secondary pulmonary nodules from non-NSCLC or NSCLC; (6) past non-NSCLC tumors with evidence of active disease at the time of SBRT and synchronous non-NSCLC tumors (arising within six months of SBRT diagnosis of NSCLC) with the exception in both cases of non-melanomatous skin tumors.

The patient cohort was divided into training (76 patients from three centers) and external validation (41 patients from the other four centers) datasets. This strategy for the distribution of centers among datasets was made to balance the two groups according to the patients' outcomes as described in the following sections. The external validation step was a fundamental part of the study in order to confirm the performances obtained in the training phase.

#### 2.1.2. SBRT Details

Conventional computed tomography (CT) simulation scans were obtained. The radiation oncologist contoured gross tumor volume (GTV) on the CT, as part of the therapeutic pathway. A 5–10 mm isotropic margin was added to GTV to generate the planning target volume (PTV). Intensity-modulated radiation therapy (IMRT) was delivered to all patients. The dose normalization ensured that at least 95% of PTV receives 100% of the prescribed dose with a homogeneous distribution. For all patients, ipsilateral and contralateral lung, heart, chest wall, esophagus, spinal cord, and bronchial trees were contoured as organs at risk (OARs).

### 2.2. Image Acquisition

All patients included in the study had PET/CT images, previously acquired as part of their care pathways, and a pre-treatment CT used for planning of SBRT. The planning

CT acquisition protocols and scanning devices differed among institutions, as reported in Table 1. PET image sets, corrected for attenuation, were acquired no more than three months before the start of the treatment. Patients fasted at least 6 h before the injection 18F-FDG tracer and the serum glucose level measured at the injection time was below 160 mg/mL in all patients. PET examinations were performed 60 min after the intravenous administration of the radiotracer using a specific protocol for each institution shown in Table 1.

**Table 1.** Protocol acquisition parameters for simulation CT and PET examinations stratified for centers. Whenever two scanners were used, a " | " indicated the different configurations.

| | Center | kV | mAs (Min–Max) | Slice Thickness (mm) | Manufacturer (s) | Convolution Kernel | Recon Diameter |
|---|---|---|---|---|---|---|---|
| | | | | CT | | | |
| TRAIN | BS | 120 | 191–401 | 3.0 | PHILIPS | B | 500 |
| | RE | 120 | 83–355 | 3.0 | GE | STD + | 500 |
| | PD | 120 | 70–363 | 2.5 | GE | BODY FILTER | 500 |
| EXT VAL | AV | 120 | 108–138 | 2.5 | PHILIPS | B | 500 |
| | NE | 120 | 40–73 | 3.0 | SIEMENS | B30f | 500 |
| | PI | 120 | 27–236 | 2.0 | SIEMENS | B30f–B31s | 500 |
| | PG | 120 | 80–200 | 2.5–3 | GE | STD + | 500 |

| | Center | Slice thickness (mm) | Manufacturer (s) | Recon diameter | Recon method |
|---|---|---|---|---|---|
| | | | PET | | |
| TRAIN | BS | 3.27 | GE | 700–815 | 3D IR/VPFXS |
| | RE | 3.27 | GE | 700–700 | 3D IR/VPFXS |
| | PD | 2–4 | PHILIPS ∣ SIEMENS | 576–815 | 3D-RAMLA/BLOB-OS-TF(PHILIPS) ∣ PSF 3i21s/(SIEMENS) |
| EXT VAL | AV | 4 | PHILIPS ∣ GE | 500–700 | BLOB-OS-TF/VPFXS |
| | NE | 2–5 | SIEMENS | 576–700 | PSF+TOF 3i21s |
| | PI | 3.27 | GE ∣ PHILIPS | 576–700 | 3D IR (GE) ∣ BLOB-OS-TF(PHILIPS) |
| | PG | 3.27 | GE ∣ SIEMENS | 600–700 | OSEM ∣ OSEM 2i8s |

*2.3. Image Segmentation*

Computed tomography and PET image sets were exported in DICOM format into a dedicated research computer for radiomics analysis. For the present study, gross tumor volume contouring was separately performed on the CT (manually, referred to as GTVCT) and PET (automatically, hereinafter named GTVPET) images of the pre-treatment PET/CT studies.

Two radiation oncologists with experience in lung cancer contoured each lesion on every sequential slice of the planning CT using standardized window settings for parenchyma (W = 1600 and L = −600), according to EORTC guidelines [31] for all patients. Regarding GTVPET delineation, the radiation oncologists placed a region of interest (ROI) on the area of tumor FDG uptake on PET images and an automatic contour—consisting of the region encompassed by a given fixed percent intensity level relative to the maximum registered tumor activity (40% of SUV max)—was generated. We decided to use this approach as a previous study showed that GTVPET delineation using this fixed threshold was better correlated with the gross tumor (based on pathologic examination) instead of using as basis the manually delineated GTVCT [32].

In order to perform radiomic feature harmonization, we used an ROI from the healthy tissue. This was obtained by copying the GTVCT into a healthy lung region, i.e., the contralateral lung at the same level of the GTV (named Contra_Lung). The Contra Lung initial volume was also shifted by 0.6 and 0.3 cm in six directions for a total of 12 shifts, avoiding the inclusion of surrounding tissues of the healthy lung. These shifts had the aim of simulating the uncertainty in the positioning of the healthy ROI (for future reproducibility of the harmonization method). We chose the shifts in accordance with PET image resolution to account for a likely uncertainty in ROI positioning since PET imaging can be used to localize the GTV before the treatment.

An example picture showing the location of the ROIs mentioned above is shown in Figure 1.

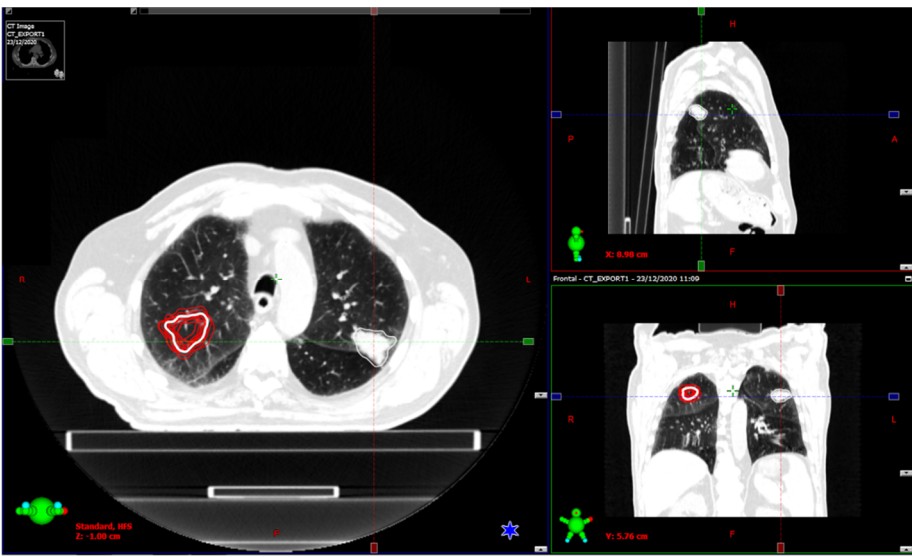

**Figure 1.** Visualization of the CT ROIs in a patient. The contralateral ROI was shifted in 12 different positions (shown in red).

*2.4. Outcome*

In this study, PFS was considered as the primary endpoint and was converted into a binary outcome, which was set to 1 for patients who were alive and without disease progression at 24 months, 0 otherwise. PFS was defined as the time from the start of the SBRT to documented relapse or death. The use of the 2-year threshold was chosen because it could properly describe the treatment effectiveness. In fact, a preliminary analysis of the Kaplan–Meier curves of PFS after SBRT for our cohort showed that the majority of the progressions occurred in a period ranging from 2 to 3 years.

*2.5. Radiomics Analysis*

Our analysis followed the steps defined for our radiomic study (Figure 2), which included image preprocessing. The first phase consisted of spatial resampling to an isotropic voxel size to obtain reproducible and rotationally invariant features. Then, image range re-segmentation updated the ROI voxels according to a chosen intensity range to remove all voxels for which intensity values fall outside the selected intensity range. Finally, the images were discretized by intensity, grouping the original intensity values (256) into specific ranges (bins). The aim was to reduce image noise and computational burden. The intensity discretization process fixed the width of the re-segmentation interval and the bin width, defining a new bin for each intensity interval. Selecting the bin width allowed direct control of the absolute range represented on each bin. The image preprocessing of intensity and spatial discretization is described in Supplementary Material Table S1. Intensity discretization parameters were chosen accordingly to the guidelines proposed by Orlhac et al. [33,34].

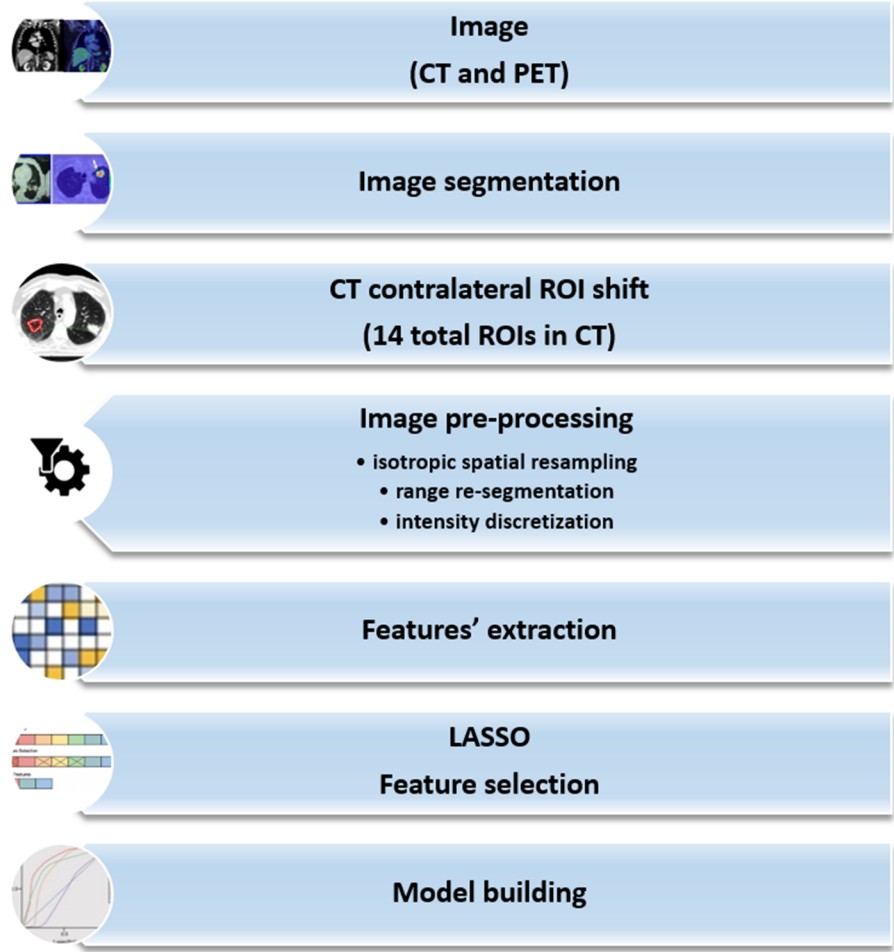

**Figure 2.** Radiomic pipeline description of the implemented steps in our evaluation process.

2.5.1. Feature Extraction

After the image preprocessing steps, feature calculation and their extraction was performed. Features (intensity-based, shape-based, and second-order) were extracted from original images and filtered images (using wavelets, Laplacian of Gaussian (LoG), and gamma modifier filters) [35]. Radiomic features were calculated using a homemade software employing the widely used pyRadiomics library in order to apply pre-determined filters to the original images and compute features from the edited results. The list of the extracted radiomic features can be found at https://pyradiomics.readthedocs.io/en/latest/features.html, (accessed on 15 July 2022).

2.5.2. Harmonization Process

The harmonization process consisted of, for our two available image modalities, calculating features for 14 different ROIs. One of them coincided with GTVCT, the other 13 with the duplicated GTVCT positioned in the healthy region and its shifts, as described in Section 2.3. This allows us to consider operators' variability in the positioning of the healthy ROI. The general idea of employing this harmonization formula was inspired by another work [36], and it is shown in Equation (1):

$$f_{HARM}(i) = \frac{f_{GTV}(i) - f_{HEALTHY}(i)}{\sigma(i)} \tag{1}$$

where: $f_{HARM}(i)$ is the ith harmonized feature, $f_{HEALTHY}(i)$ is the median ith feature value calculated on the 13 healthy tissue samples for each patient and modality, and $\sigma(i)$ is the

difference between the 75th and the 25th percentile of the i[th] feature distribution. Shape features were not harmonized. We applied this harmonization only to CT data because of its intrinsic dependence on the protocol acquisition parameters. Furthermore, CT is used for a morphologic and anatomical characterization and pixel values are related to a physical characteristic of the tissue (the linear attenuation coefficient). On the other hand, PET, being a functional imaging modality, is less sensitive to low signal changes in spatial coordinates. Especially in this case, for the healthy lung, in PET pixel values, there is no useful physical information regarding a region where we do not register a signal from the radiotracer absorption.

### 2.5.3. Feature Selection

LASSO feature selection was applied, in which the following function is minimized (Equation (2))

$$\sum_{i=1}^{n}\left(y_i - \beta_0 - \sum_{j=1}^{p}\beta_j x_{ij}\right)^2 + \lambda\sum_{j=1}^{p}\left|\beta_j\right| \tag{2}$$

where: $y_i$ is the observed value, $\beta_j$ is the LASSO coefficients, and $\lambda\sum_{j=1}^{p}\left|\beta_j\right|$ is the shrinkage penalty [37].

The parameter $\lambda$ was chosen using 10-fold cross-validation (CV) computing its error. LASSO penalty brings to zero the weight coefficients ($\beta_j$) of irrelevant features not predictive of the chosen outcome. In addition, LASSO handles sets of collinear features by increasing the weight of one of them while setting the other weights to zero. Because the considered outcome was binary, we used a binomial function for LASSO regression. In Table S2 we show the shrinkage penalties for our trained models.

### 2.5.4. Model Building

The original and harmonized features were used to develop a supervised machine learning binary classifier. A linear support vector machine (SVM, Model 1) [38] and an Ensemble Subspace Discriminant (ESD, Model 2) [39] were trained by optimizing their performance in 10-fold cross-validation in the training dataset.

Linear SVM classifiers provide low generalization error, even with small learning sample datasets. ESD classifiers are used to decide an explicit discriminant subspace of low dimension.

The two described model types were applied to five different combinations of input features: (A) harmonized CT + PET, (B) harmonized CT, (C) original CT, (D) only original PET, and (E) harmonized CT + PET + selected clinical variables in order to assess the effect of harmonization on the performance of the predictive models. The interested reader can find more information in Text S1 in Supplementary Materials.

The clinical variables in method (E) were chosen among the available ones by using Kaplan–Meier survival curves as described in Section 2.5.5.

### 2.5.5. Statistical Analysis

For each model, a 95% confidence interval (CI) of the AUCs was calculated for the training and external validation sets. Furthermore, accuracy (95% CI are reported), precision, and recall were calculated.

Subsequently, Kaplan–Meier survival curves were computed using the PFS to select the clinical features. A clinical feature exhibiting a *p*-value from a log-rank test less than 0.05 was considered significant and included in model E. Matlab R2021b (Mathworks, Natick, MA) and R (Vienna, Austria), available at https://www.R-project.org (accessed on 15 July 2022), were used to perform the statistical analysis.

The *p*-values related to statistical differences among the AUC values of each model were calculated using two-sided DeLong test.

## 3. Results

### 3.1. Clinical Results

One-hundred and seventeen early-stage NSCLC patients met the inclusion criteria. The baseline characteristics of the patients are summarized in Table 2. The median age was 78 years and there were more male (72.6%) than female patients. With a median follow-up of 29.8 months, the median PFS was 24.2 months, and 2-year PFS percentage was 51.2%. Median OS and 2-year OS percentage were 28.5 months and 64%, respectively. The clinical characteristics, including age, gender, Charlson comorbidity index (CCI), diffusing capacity of carbon monoxide (DLCO), tumor size, Eastern Cooperative Oncology Group (ECOG) performance status, and biological equivalent dose to PTV, showed no significant differences between the training and external validation cohorts (Table 2).

**Table 2.** Statical analysis of clinical variables. Abbreviations: PS: Performance status according to ECOG scale, BPCO: chronic obstructive pulmonary disease; ADK: Adenocarcinoma, SCC: squamous cell carcinoma, Fr: fraction; RT: radiotherapy VMAT: volumetric arc-therapy; IMRT: intensity modulated radiotherapy, TOMO: Tomotherapy, PTV: planning target volume. *p*-values in bold mean the statistical significance.

| Characteristics | Training Cohort (N = 76) | External Validation Co#Hort (N = 41) | *p* |
|---|---|---|---|
| Gender | | | |
| Male | 61 | 24 | |
| Female | 15 | 17 | **0.04** |
| Age (years) | 78 [51–87] | 79 [57–88] | 0.72 |
| Smoking Status | | | |
| Yes | 50 | 27 | |
| No | 26 | 14 | 0.22 |
| Performance Status | | | |
| 0 | 37 | 18 | |
| 1 | 35 | 15 | 0.75 |
| 2 | 4 | 7 | |
| BMI | 25.2 [16.4–37.1] | 24.8 [18.3–44.7] | 0.17 |
| Diabetes mellitus | | | |
| Yes | 16 | 12 | |
| No | 60 | 29 | 0.58 |
| BPCO | | | |
| Yes | 43 | 17 | |
| No | 19 | 24 | 0.54 |
| Charlson Comorbidity Index (CCI) | | | |
| Median | 6.5 | 6 | |
| Range | [3–13] | [4–10] | 0.55 |
| T diameter | | | |
| Median | 2.35 | 2.3 | |
| Range | [0.6–5.5] | [0.72–27] | 0.58 |
| Lesion type | | | |
| Subsolid | 5 | 4 | |
| Solid | 71 | 37 | 0.42 |
| Lung Side | | | |
| Lung right | 42 | 22 | |
| Lung left | 34 | 19 | **0.006** |

**Table 2.** *Cont.*

| Characteristics | Training Cohort (N = 76) | External Validation Co#Hort (N = 41) | p |
|---|---|---|---|
| Lobe Site | | | |
| Upper Lobe | 44 | 23 | |
| Lower Lobe | 30 | 15 | 0.89 |
| Middle Lobe | 2 | 1 | |
| Lesion Site | | | |
| Peripheral | 55 | 34 | |
| Central | 21 | 7 | 0.92 |
| $BED_{10}$ | | | |
| Median | 115.5 | 100 | |
| Range | [100–180] | [100–132] | 0.64 |

No clinical or treatment-related features were shown to be significantly related to PFS in the univariate analysis of the whole population, except for gender ($p = 0.04$ in favor of female) and lung site (right vs. left in favor of the right one, $p = 0.006$).

### 3.2. PFS Models

The PFS predictive performance of the linear SVM and ESD models using radiomic features and clinical features are reported in Table 3. In Figure 3, all the models are graphically compared considering their confidential intervals. Models using harmonized features and PET (A,E) achieved AUCs greater than 0.70, both in training and validation. The performances of models using CT-only harmonized features (B models) are not confirmed on the validation dataset (AUC training > 0.75, AUC validation <0.60), while adding PET features leads to better stability between the training and validation sets. Only C-type models (original CT-only features) showed a low-mean AUC (< 0.62), both in training and validation.

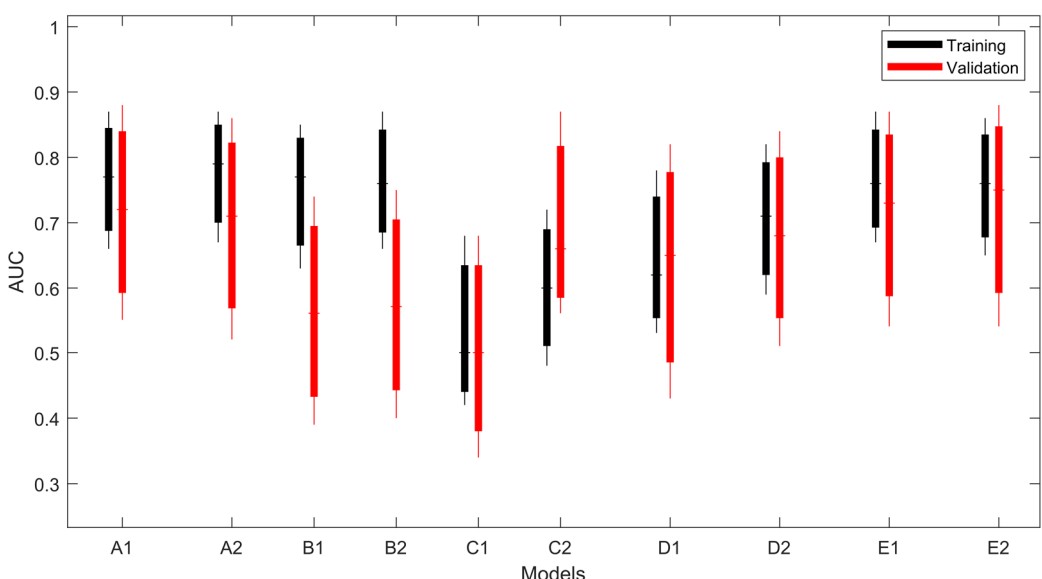

**Figure 3.** Performances (AUC) of the studied models. The boxplot shows the minimum, maximum, and average values of the bootstrapped 95% CIs.

**Table 3.** Models' results in terms of AUC, accuracy, precision, and recall.

| | Harmo CT + Original PET Features (A) | | | | |
|---|---|---|---|---|---|
| | **Linear SVM (A1)** | | | | |
| | AUC * | Accuracy | Precision ** | Recall ** | *p* *** |
| Training dataset | 0.77 [0.66–0.87] | 0.72 ± 0.02 | 0.67 | 0.83 | **1.0 × 10⁻⁴** |
| External validation dataset | 0.75 [0.55–0.88] | 0.66 ± 0.01 | 0.68 | 0.65 | **0.01** |
| | Subspace Discriminant (A2) | | | | |
| | AUC * | Accuracy | Precision ** | Recall ** | *p* *** |
| Training dataset | 0.79 [0.67–0.87] | 0.71 ± 0.01 | 0.69 | 0.83 | **0.02** |
| External validation dataset | 0.71 [0.52–0.86] | 0.63 ± 0.02 | 0.68 | 0.65 | **0.046** |
| | Harmo CT features (B) | | | | |
| | Linear SVM (B1) | | | | |
| | AUC | Accuracy | Precision ** | Recall ** | *p* *** |
| Training dataset | 0.77 [0.63–0.85] | 0.67 ± 0.02 | 0.74 | 0.58 | **1.0 × 10⁻⁴** |
| External validation dataset | 0.56 [0.39–0.74] | 0.58 ± 0.01 | 0.67 | 0.52 | 0.5 |
| | Subspace Discriminant (B2) | | | | |
| | AUC | Accuracy | Precision ** | Recall ** | *p* *** |
| Training dataset | 0.76 [0.66–0.87] | 0.71 ± 0.02 | 0.73 | 0.6 | **0.01** |
| External validation dataset | 0.57 [0.4–0.75] | 0.58 ± 0.01 | 0.67 | 0.52 | 0.50 |
| | Original CT features (C) | | | | |
| | Linear SVM (C1) | | | | |
| | AUC | Accuracy | Precision ** | Recall ** | |
| Training dataset | 0.56 [0.42–0.68] | 0.52 ± 0.03 | 0.49 | 0.45 | |
| External validation dataset | 0.50 [0.34–0.68] | 0.43 ± 0.02 | 0.54 | 0.65 | |
| | Subspace Discriminant (C2) | | | | |
| | AUC | Accuracy | Precision ** | Recall ** | |
| Training dataset | 0.63 [0.48–0.72] | 0.56 ± 0.03 | 0.58 | 0.56 | |
| External validation dataset | 0.51 [0.39–0.74] | 0.54 ± 0.01 | 0.58 | 0.65 | |
| | PET features only (D) | | | | |
| | Linear SVM (D1) | | | | |
| | AUC | Accuracy | Precision ** | Recall ** | *p* *** |
| Training dataset | 0.68 [0.53-0.78] | 0.64 ± 0.03 | 0.64 | 0.80 | 0.09 |
| External validation dataset | 0.65 [0.43-0.82] | 0.64 ± 0.01 | 0.67 | 0.78 | 0.18 |
| | Subspace Discriminant (D2) | | | | |
| | AUC | Accuracy | Precision ** | Recall ** | *p* *** |
| Training dataset | 0.71 [0.59–0.82] | 0.69 ± 0.01 | 0.67 | 0.8 | 0.10 |
| External validation dataset | 0.68 [0.51–0.84] | 0.60 ± 0.01 | 0.67 | 0.61 | 0.08 |
| | Harmo CT + Original PET + Clinical features (E) | | | | |
| | Linear SVM (E1) | | | | |
| | AUC * | Accuracy | Precision ** | Recall ** | *p* *** |
| Training dataset | 0.79 [0.67–0.87] | 0.73 ± 0.02 | 0.72 | 0.83 | **6.0 × 10⁻⁵** |
| External validation dataset | 0.73 [0.54–0.87] | 0.73 ± 0.01 | 0.77 | 0.74 | **0.02** |
| | Subspace Discriminant (E2) | | | | |
| | AUC * | Accuracy | Precision ** | Recall ** | *p* *** |
| Training dataset | 0.76 [0.65–0.86] | 0.74 ± 0.01 | 0.72 | 0.83 | 0.01 |
| External validation dataset | 0.75 [0.54–0.88] | 0.68 ± 0.02 | 0.73 | 0.70 | 0.02 |

* AUCs in square brackets are their bootstrapped 95% CIs. ** Precision and recall are presented for class 1. *** *p*-values are calculated with respect to the conditions C1 and C2 for linear SVM and ESD models, respectively. Values in bold mean the statistical significance.

It is worth noting that both A1 and A2 models significantly outperformed C1 and C2 models, both in the training and external validation datasets ($p = 0.0001$, $p = 0.01$ and $p = 0.02$, $p = 0.046$, respectively, for linear SVM and subspace discriminant models), likewise for E and C models (E1: $p < 0.0001$, and $p = 0.02$, and E2: $p = 0.01$, and $p = 0.02$, respectively, for training and external validation datasets). C models outperformed B models, but only in the training dataset ($p < 0.0001$ and $p = 0.01$, respectively, for linear SVM and subspace

discriminant models) and D and C models had the same performance metrics. In summary, models using clinical information (E models) do not add a significant improvement to A models. This effect is also appreciable in Figure 4, where all the *p*-values among the models are represented.

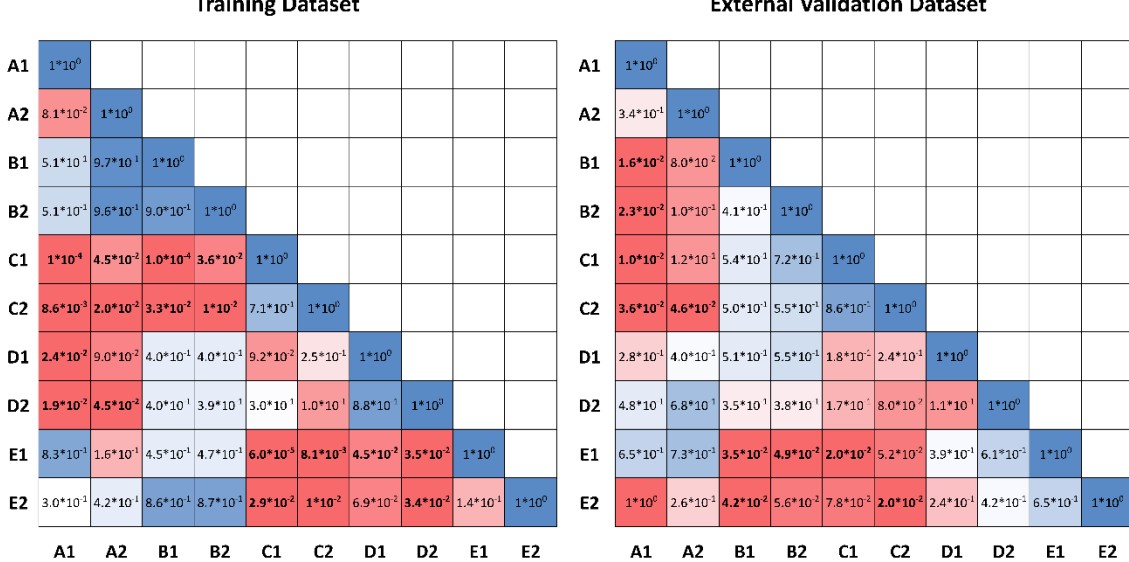

**Figure 4.** *p*-values calculated using the two-sided DeLong test. Numbers in bold mean the statistical significance.

## 4. Discussion

In this work, a multi-centric cohort of early-stage NSCLC patients treated with SBRT was used to build and validate predictive models of PFS greater than 24 months using radiomic features from CT and PET exams and clinical information. Several existing works in the literature [40] describe acquisition protocol variabilities in multi-centric studies, which could affect the performance of radiomic models. Since radiomics computes features from the pixel values in the images, differences in acquisition protocol can lead to biased results. The rationale behind our harmonization method lies in the fact that retrospective multicenter radiomic studies are challenging but necessary, as gathering data from several centers for a centralized analysis is complex for legal, ethical, administrative, and technical reasons. Most of the time, the different centers involved do not follow standardized acquisition and reconstruction protocols; therefore, the collected data suffer from intra-, and inter-variability, making radiomic features sensitive to multicenter variability. Our novel harmonization technique aims to reduce the bias caused by the absence of standardized protocols. Generally, feature analysis is performed by calculating them inside an ROI that coincides with the lesion target. Our study aims to tackle this issue by attempting to reduce this effect using the healthy region of the patient as the baseline from which to harmonize the radiomic data computed from the lesion. From our results, the harmonization improves models' performance when it is used on CT image sets. On the other hand, we expected that for PET-only images, the harmonization method is not easily applicable due to the functional aim of this imaging modality. In such a modality, several healthy tissues (i.e., lungs) are not 18-FDG-avid, while a harmonization based on healthy radiotracer accumulation, to our knowledge, has yet to be studied. Our work investigated and evaluated the feasibility of this technique, which could be employed and better analyzed in future studies.

When using original feature values, PET features were preferred over CT during feature selection, resulting in an only PET-based model. Furthermore, in A models, two features from CT were included in the final prediction score (Log_Sigma30mm_GLDM_Small-Dependence-High-Gray-Level-Emphasis (SDHGLE) and Wavelet_LHH_NGTDM_Busyness), which were also selected in B models. In the same way, a subset of selected features in the A

method (Log_sigma10mm_GLSZM_Size-Zone-Low-Gray-Level-Zone-Emphasis (LGLZE), Exponential_FIRST ORDER_Median, Square_GLSZM_ZoneEntropy (ZE)) is also present among the selected features in the PET-only-based method. This could mean that our approach, also given the higher performance metric of the A method, was able to merge the information hidden in the CT and PET image sets in a multi-centric cohort of patients, highlighting the importance of properly handling hybrid imaging in radiomic models.

Other harmonization techniques were previously described in the literature [41]. Among these, the ComBat harmonization method, which removes batch effects, mostly based on an empirical Bayes framework, is one of the most used. Conversely, the ComBat method has some limitations: for instance, the dimension of the homogeneous group cannot be too few (in our study, it would have not been applicable as most of the centers provided less than 15 patients). Indeed, recently, some methods have been proposed to overcome these limitations, e.g., using bootstrap and Monte Carlo technique to improve robustness in the estimation [42].

Even if Monte Carlo and bootstrap strategies aimed at overcoming the cohort size limitations, the objective of this method still focuses on removing differences in radiomic feature distributions among different labels (corresponding to different centers). ComBat, thus, relies on the individual distributions and changes made to feature values are dependent on a group of patients. While we know that there are data supporting the effectiveness of this method (especially in making the feature distributions uniform), we wanted to tackle the multicenter studies issue from a different angle, which is to account for the individual patients' differences (caused both by the scanner/institution protocols and their anatomy) taken directly from the lesion imaging. This renders the method easier, both computationally and for cohort eligibility reasons (which, in the ComBat method, is needed for representing the single center in terms of homogeneity being an assumption of the method). In fact, if validated further, our method can be applied even in heterogenous cohorts since it uses only the single image set of the patient.

Our approach aimed to use all the information present in the CT data, both from cancer lesions and the contralateral healthy tissue, simulating the radiologists' skill in subjectively evaluating a lesion and adding this information in quantifiable and statistical terms (through the radiomic features).

In our work, the well-known and studied concept of delta radiomics was implemented not in a temporal sense but spatially (cancer vs. healthy tissue), which is an approach that, to our knowledge, was not applied in other prognostic oncological works. Traditional radiomics uses absolute values extracted from regions of interest to predict a clinical outcome. On the other hand, delta radiomics predicts a clinical outcome through the combination of radiomic values computed from image sets acquired at different time points (i.e., radiographs to monitor follow-ups or differences between basal PET and interim PET), which is a rationale also used in clinical practice to assess lesion progression (i.e., PERCIST). In our manuscript, we decided to apply delta radiomics not between different time points but between different anatomic locations (healthy vs. tumor tissues). The assumption behind this use of delta radiomics is that each patient can have an intrinsic "baseline" value for a certain radiomic feature (caused by individual anatomy and institution protocols) that needs to be accounted for when building predictive models. Comparison between normal and tumor tissue behavior (even in terms of pixel values) is also common in clinical practice (i.e., SUV values typical of physiological metabolism or HU/density values of healthy tissue). Some authors [43–45] explain that delta radiomics—which is the use of textural indices associated with different time points or anatomical regions—is more successful than traditional radiomics. Our work aims to provide the basis for a framework where the study of simple absolute feature values can make room for the analysis of their relationship to a reference, used as a threshold or as a comparison.

There are several limitations to the current work. Our study suffered a restricted number of patients selected retrospectively. Nonetheless, the patients' number seemed reasonable at the current phase of our study. It assures the homogeneity in terms of patholo-

gies, as including only NSCLC lesions prevented possible biases created by evaluating different diseases, even in the lung anatomical district. Our study highlighted the necessity to monitor and carefully use features related to pixel values and their relationships. In this case, we can assume that the importance of the radiomic features is not only held in their numerical value but also in the intrinsic relationship among those values. The textural indices' ability to perform more complex clinical tasks (i.e., predicting toxicity and its grade) could be further examined in the next phase of our work or in a prospective study design, which could also assess the robustness of our method. We believe that a prospective study will be able to validate these models within a cohort gathered with a better strategy.

Interestingly enough, we found that the improved performance in models employing harmonized features in the training phase was also confirmed in the validation dataset; the use of an external dataset is becoming more and more crucial to radiomics studies to assure and facilitate their introduction in clinical practice.

In our study, no clinical or treatment-related features were shown to be significantly related to PFS, except for gender and lung site. It is well known in the literature that gender is a prognostic factor for PFS [46–48]. Due to the size of our cohort, we did not find significant correlations between PFS and other studied clinical prognostic variables, such as age or histology (also due to the inclusion criteria). To our knowledge, we did not find any other study reporting a significant correlation between PFS and tumor laterality; thus, we will investigate this finding together with our model generalization power in a future prospective study. Indeed, in the literature, many studies showed a significant correlation between some clinical or treatment-related characteristics and outcomes (PFS and OS) and some created predictive models. Among the various statistical prediction models, nomograms can be accurate and feasible prognostic instruments with high utility in estimating individual patient risk and may, thus, help guide treatment decisions in clinical practice. At present, there are some nomograms, based on clinical features, developed for early-stage NSCLC treated with SBRT [49–51], but there is still need for validation of the clinical variables found in those studies and their experimental results in more robust cohorts, such as prospective ones. Therefore, a need exists for a robust recurrence-related prediction model to help select high-risk candidates who may benefit from additional systemic therapies.

In this scenario, a predictive model based on radiomics and clinical and treatment-related characteristics can improve the prediction of clinical outcomes, as already demonstrated by other works. We also found out that clinical variables did not improve the radiomics models, but only the proposed harmonization process statistically significantly improves the model's performance.

As previously stated, our future aim is to apply our method to a prospective multi-centric cohort to further validate the framework's stability. In addition, other anatomical regions should be explored to generalize the harmonization, even when the healthy area is not so easily defined as in the lung case. Regarding the employment of this method also in PET datasets, it could be interesting to explore the feasibility of applying our harmonization to 18F-FDG-avid anatomical regions, such as the liver or the brain, which are, however, not related to a pathologic response. Such a method could be especially useful where CT-PET is the only exam included in the care pathway of the patient.

## 5. Conclusions

A novel strategy of CT data harmonization involving delta radiomics, considering both cancer and healthy tissue in the contralateral lung, was tested and externally validated in a multi-centric study for NSCLC patients, to initially assess its feasibility.

The radiomics models with harmonized features can predict better the selected patient outcome in our cohort, providing valuable additional information to the clinician.

**Supplementary Materials:** The following are available online at https://www.mdpi.com/article/10.3390/curroncol29080410/s1, Table S1: The image preprocessing of intensity and spatial discretization, Table S2: Lasso feature selection, Text S1: Models' description.

**Author Contributions:** M.B.: study, design, literature search, data analysis/interpretation, writing original draft, supervision, V.T.: study, design, literature search, data analysis/interpretation, writing original draft, A.B.: Data analysis/interpretation, data validation, reviewing draft, N.C.: literature search, data analysis/interpretation, data validation, figures, writing, M.G.: study design, data collection, S.C.: literature search, reviewing draft, P.B.: resources, data collection, S.L.M.: resources, data collection, E.P.: resources, data collection, M.A.: resources, data collection, reviewing draft, A.R.: resources, data collection, M.S.: resources, data collection, C.P.: resources, data collection, S.U.: resources, data collection, G.M.: resources, data collection, N.G.-L.: resources, data collection, L.F.: resources, data collection, C.I.: resources, data collection, M.I.: resources, data collection, P.C.: resources, data collection, supervision. All authors have read and agreed to the published version of the manuscript.

**Funding:** This study was partially supported by the Italian Ministry of Health—Ricerca Corrente.

**Institutional Review Board Statement:** The study was conducted in accordance with the Declaration of Helsinki, and approved by the Ethics Committee of by the Area Vasta Emilia Nord (AVEN) (protocol code 817/2018/OSS*/IRCCSRE approved on 16 July 2019).

**Informed Consent Statement:** Informed consent was obtained from all subjects involved in the study.

**Data Availability Statement:** The training weights of the models proposed in this work (namely, models E and A described in Section 2.5.4) are available in a GitHub repository at the link: https://github.com/ausl-re/TEXAS, accessed on 15 July 2022. Further instructions on how to perform the harmonization and to use the model will be added.

**Acknowledgments:** A heartfelt thanks to all the people involved in the various centers in collecting and managing the data, especially Simona Marani, Maria Paola Ruggieri, Giulia Mascari, and Cinthia Aristei. A special thanks to the AIRO Lung Group, especially Vieri Scotti, Stefano Vagge, and Alessio Bruni.

**Conflicts of Interest:** The authors declare no conflict of interest.

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
