# Peer review of "Novel Harmonization Method for Multi-Centric Radiomic Studies in Non-Small Cell Lung Cancer"

_curroncol, doi:10.3390/curroncol29080410_

Round 1

Reviewer 1 Report

This multicenter study investigated the effectiveness of the harmonizing CT to develop PFS predicting models using radiomics features from CT and PET for NSCLC patients treated with SBRT. The models using the harmonizing CT showed better performance to predict PFS than models not using that. The authors described that the harmonizing technique could improve AI model performance by reducing bias from image acquisition. Because SBRT and AI models predicting prognosis has been emerged recently, this study may have interests of readers.

Author Response

Referee 1

Comments and Suggestions for Authors

This multicenter study investigated the effectiveness of the harmonizing CT to develop PFS predicting models using radiomics features from CT and PET for NSCLC patients treated with SBRT. The models using the harmonizing CT showed better performance to predict PFS than models not using that. The authors described that the harmonizing technique could improve AI model performance by reducing bias from image acquisition. Because SBRT and AI models predicting prognosis has been emerged recently, this study may have interests of readers.

R: We thank the Referee for their observations.

Reviewer 2 Report

This paper proposed a new, simple and effective harmonization technique to extract radiomics features from lesion and health areas. The authors used LASSO to select features, and 2 models to predict 24-months progression-free-survival.

This method makes sense to me. Although the sample data is small, this is a multi-center study and used an external validation set. Therefore, the generalizability of the prediction model is considered good. However, the logic of the article is not rigorous enough, especially the discussion part needs to be further improved.

Here are some specific comments:

1. The evaluation of the improved performance is not clear. What is the method used to calculate the p-value in Table 3? deLong test or likelihood ratio test? It would be interesting to see if there are any increment values between model A and B, A and D, A and E, respectively. 

2. What are the differences in the distributions of clinical variables for predicting PFS compared to other studies? As a multi-center database, the reasons for the different distribution from other data deserve discussion. Is there any other data with a similar scenario?

3. Authors should provide detailed data or experimental flaws to support their statement that “… there are still uncertainties about their applicability and robustness” in [44-46].

4. What are the advantages of the authors’ harmonization techniques compared to other good harmonization techniques mentioned in the Discussion (e.g. Monte Carlo, bootstrap). 

5. The definition of delta-radiomics/traditional radiomics is not clear. 

6. It may be a better choice to show the differences between different models with Figures.

Author Response

Referee 2

Comments and Suggestions for Authors

This paper proposed a new, simple and effective harmonization technique to extract radiomics features from lesion and health areas. The authors used LASSO to select features, and 2 models to predict 24-months progression-free-survival.

This method makes sense to me. Although the sample data is small, this is a multi-center study and used an external validation set. Therefore, the generalizability of the prediction model is considered good. However, the logic of the article is not rigorous enough, especially the discussion part needs to be further improved.

R: We thank the Referee for their comments. We added the Referee’s proposals to the manuscript and regarding the models performance, we decided to report the mean AUC value computed from the bootstrap. We completed the statistical analysis and comparisons of the built models also adding comparisons also between A and B, D, E models. We think that the paper significantly improved thanks to the Referee’s observations, which were useful to support our initial conclusions.

Here are some specific comments:

  1. The evaluation of the improved performance is not clear. What is the method used to calculate the p-value in Table 3? deLong test or likelihood ratio test? It would be interesting to see if there are any increment values between model A and B, A and D, A and E, respectively.

R: We thank the Referee for the observation. We used deLong 2-tailed test to compute the p-values and we added, as suggested, also Figure 4 where we report the p-values for all combinations of our analysed models. The shown table is symmetric so we reported only one set of values. From these results, we obtained that A model performance remains stable both in training and validation and thus differences with B models are found only in the latter case. A models are superior to D models only in the training dataset, and A and E models performances are not statistically significant, meaning that in our cohort we could not quantify the potential power of combining radiomic with clinical prognostic factors.

We added the following sentence to the manuscript: “We also obtained that A model performance remains stable both in training and validation and thus differences with B models are found only in the latter case. A models are superior to D models only in the training dataset, and A and E models performances are not statistically significant. This effect is also appreciable in Figure 4 where all the p-values among the models were represented.”

  1. What are the differences in the distributions of clinical variables for predicting PFS compared to other studies? As a multi-center database, the reasons for the different distribution from other data deserve discussion. Is there any other data with a similar scenario?

R: We added the following sentence and references to the manuscript Discussion:

“It is well-known in literature that gender is a prognostic factor for PFS. [50-52] Due to the size of our cohort, we did not find significant correlations between PFS and other studied clinical prognostic variables such as age or histology (also due to the inclusion criteria). To our knowledge, we did not find any other study reporting a significant correlation between PFS and tumor laterality; thus, we will investigate this finding together with our model generalization power in a future prospective study.”

50 Pinto JA et al. Gender and outcomes in non-small cell lung cancer: an old prognostic variable comes back for targeted therapy and immunotherapy? ESMO Open 2018, 3, e000344

51 de Perrot et al. Sex differences in presentation, management, and prognosis of patients with non-small cell lung carcinoma. J Thorac Cardiovasc Surg 2000, 119, 21–6

52 Hsu LH et al. Sex-associated differences in non-small cell lung cancer in the new era: is gender an in-dependent prognostic factor? Lung Cancer 2009, 66, 262–7

  1. Authors should provide detailed data or experimental flaws to support their statement that “… there are still uncertainties about their applicability and robustness” in [44-46].

R: We thank you for your suggestion. The nomograms highlighted in the referenced studies take into consideration different clinical variables. Thus, there is not a validated group of clinical variables which can be used in a standardized nomogram.

We changed the sentence in “ …there is still need for validation of the clinical variables found in those studies and their experimental results in more robust cohorts, such as prospective ones.”

  1. What are the advantages of the authors’ harmonization techniques compared to other good harmonization techniques mentioned in the Discussion (e.g. Monte Carlo, bootstrap).

R: In the discussion section, we provided a description of our method’s advantages compared to ComBat method: “Among these, the ComBat harmonization method, which removes batch effects mostly based on an empirical Bayes framework, is one of the most used. Conversely, ComBat method has some limits: for instance, the dimension of the homogeneous group cannot be too few (in our study it would have not been applicable as most of the centers provided less than 15 patients). Indeed, recently some methods have been proposed to overcome these limitations, e.g., using bootstrap and Monte Carlo technique to improve robustness in the estimation”.

To enhance the advantages of our model with respect to a statistical approach we added the following sentence to the manuscript: “Even if Monte Carlo and bootstrap strategies applied in ComBat aimed at overcoming the cohort size limitations, the objective of this method, still focuses on removing differences in radiomic features distributions among different labels (corresponding to different centres). ComBat thus relies on the individual distributions and changes made to feature values are dependent on a group of patients. While we know that there is data supporting the effectiveness of this method (especially in making the feature distributions uniform), we wanted to tackle the multicenter studies issue from a different angle, which is to account for the individual patients’ differences (caused both by the scanner/institution protocols and their anatomy) taken directly from the lesion imaging. This renders the method easier both computationally and for cohort eligibility reasons (which, in ComBat method, is needed for representing the single center in terms of homogeneity being an assumption of the method). In fact, if validated further, our method can be applied even in heterogeneous cohorts since it uses only the single image set of the patient.”   

  1. The definition of delta-radiomics/traditional radiomics is not clear.

R: We added the following sentence to the manuscript: “Traditional radiomic uses absolute values extracted from regions of interest to predict a clinical outcome. On the other hand, delta radiomics predicts a clinical outcome through the combination of radiomic values computed from image sets acquired at different time points (i.e. radiographs to monitor follow-ups or differences between basal PET and interim PET), which is a rationale also used in clinical practice to assess lesion progression (i.e. PERCIST). In our manuscript, we decided to apply delta radiomics not between different time points but between different anatomic locations (healthy vs tumor tissues). The assumption behind this use of delta radiomics is that each patient can have an intrinsic “baseline” value for a certain radiomic feature (caused by individual anatomy and institution protocols) that needs to be accounted for when building predictive models. Comparison between normal and tumor tissue behavior (even in terms of pixel values) are also common in clinical practice (i.e. SUV values typical of physiological metabolism or HU/density values of healthy tissue).”

  1. It may be a better choice to show the differences between different models with Figures.

R: We thank you for your observation. We added Figure 3 to the manuscript showing the different AUC values of the studied predictive models.

Round 2

Reviewer 2 Report

The authors have addressed all of my comments and concerns in the revised version. I have no additional comments. Overall, the manuscript is well written and the reported results are of valuable interest to readers. I recommend accepting the paper.